# Softmax Output Approximation for Activation Memory-Efficient Training of Attention-based Networks

**Changhyeon Lee and Seulki Lee**
UNIST (Ulsan National Institute of Science and Technology)
{changhyeon,seulki.lee}@unist.ac.kr

## Abstract

In this paper, we propose to approximate the softmax output, which is the key product of the attention mechanism, to reduce its activation memory usage when training attention-based networks (aka Transformers). During the forward pass of the network, the proposed softmax output approximation method stores only a small fraction of the entire softmax output required for back-propagation and evicts the rest of the softmax output from memory. Then, during the backward pass, the evicted softmax activation output is approximated to compose the gradient to perform back-propagation for model training. Considering most attention-based models heavily rely on the softmax-based attention module that usually takes one of the biggest portions of the network, approximating the softmax activation output can be a simple yet effective way to decrease the training memory requirement of many attention-based networks. The experiment with various attention-based models and relevant tasks, i.e., machine translation, text classification, and sentiment analysis, shows that it curtails the activation memory usage of the softmax-based attention module by up to 84% (6.2× less memory) in model training while achieving comparable or better performance, e.g., up to 5.4% higher classification accuracy.

## 1 Introduction

Recently, many attention-based networks, aka Transformers [50], are widely used in many natural language processing tasks such as machine translation [50, 13], sentiment analysis [52, 29, 34], text classification [52, 54], and question/answering [47, 53, 20]. Due to their superior performance, they also have become the backbone of many large language and general task-agnostic models, e.g., GPT series [41, 5], and are actively expanding to other domains ranging from computer vision (Vision Transformers [12, 49, 38]) to protein structure predictions (AlphaFold [44, 22]). However, the stellar performance of attention-based networks comes with significant resource requirements in training, especially massive memory (RAM) usage. For example, the small-size GPT-2 model [41] is known to require 27.5 GB of peak memory for its training with a mini-batch of size eight [3]. Hence, the memory bottleneck problem makes training of attention-based models challenging when massive memory capacity is not available, e.g., BERT [11] takes 16 TPUs [21], each equipped with 32 TB of built-in memory, for training, and the regular GPT-2 [41] is reported to be trained with 32 TPUs.

During deep model training, it is required to store various components in memory, i.e., model parameters, input data, optimizer state, etc., along with activation output, i.e., the intermediate output of each layer. As the activation output of each layer must be stored in memory for gradient-based back-propagation [42], the memory capacity required to train a model increases proportionally to the data sequence length (width), depth of the network, and mini-batch size of the model. Consequently, as the model gets huge and the training data gets bigger, the activation easily becomes to take up more memory than other components. Especially, it is more evident in NLP (natural language processing) tasks, as provided in Figure 1(a) [25], which shows the memory composition required for training four different attention-based model configurations, ranging from 22 billion to 1 trillion parameters.

37th Conference on Neural Information Processing Systems (NeurIPS 2023).

For GPT-3 [5] that has 175B parameters, the activation output requires approximately $1.45\times$ more memory than the parameter and optimizer state, and for MT-NLG [45] that has 530B parameters, it is about $3.6\times$ more. Therefore, to popularize such large models in practice, it is necessary to reduce the massive amount of activation memory required in model training.

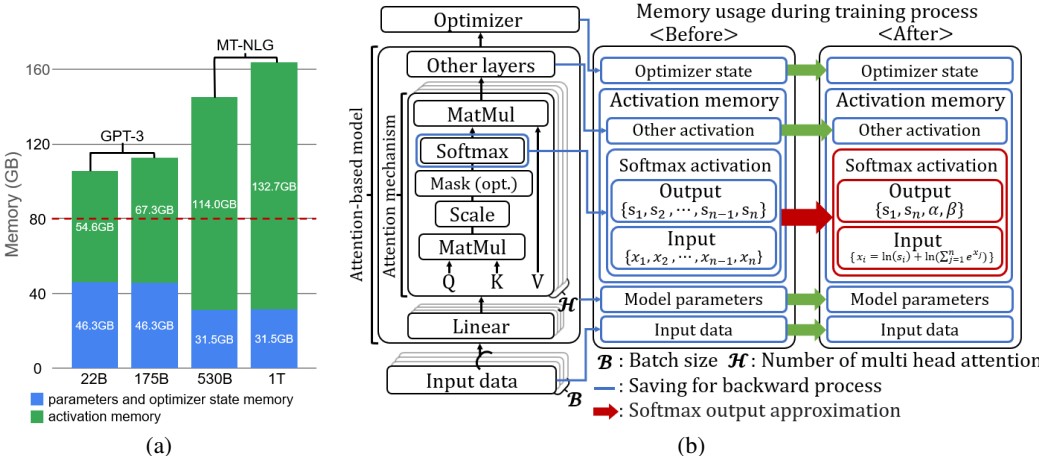

(a)  (b)

Figure 1: (a) The amount of memory required by 1) parameters and optimizer state and 2) activation memory [25] in training attention-based models (GPT-3 [5] with 22B and 175B parameters and MT-NLG [45] with 530B and 1T parameters). The dashed red line represents 80GB of memory capacity available in NVIDIA A100 GPU [35]. (b) The detailed memory composition when training an attention-based network (blue boxes). The softmax output approximation reduces the activation memory usage by storing only a small subset of them (e.g. $s_1, s_n, \alpha, \beta$) in memory (red boxes).

In this paper, we propose an approximation method of the softmax output [4] produced in the attention module [50] to reduce its activation memory requirement in model training, as shown in Figure 1(b). Instead of storing the entire elements of the softmax output vector that represents attention scores, it stores only a small subset of the elements and evicts the rest from memory during the forward pass of the network. Later, during the backward pass, the discarded softmax output elements are approximated to construct the gradients for back-propagation. As only a small fraction of the softmax activation output is stored during the forward pass, a substantial amount of memory can be saved. The proposed method is not to efficienate the computation complexity of the attention module, such as [9, 48, 31, 51], but to reduce the amount of activation memory used in training by storing only selected parts of the attention score, i.e., softmax output, in memory, not all of them. Thus, it is applicable to a softmax output computed from any form of the attention module [9, 48, 31, 51] in a model-agnostic way. To the best of our knowledge, this is the first and unique attempt to curtail the activation memory usage of the attention module in training by approximating the softmax output.

The efficacy of approximating the softmax output is substantial because it is the key component of the attention block used to compute the normalized attention score [48]. Since many attention-based models repeatedly use multi-head attentions in their encoders and/or decoders [50], the softmax output can be accounted as the dominant activation in the network. For instance, the softmax operation, i.e., $\text{Softmax}(QK^T/\sqrt{d_k})$, takes up 80% of the attention module itself, and 64.7% and 62.2% of the entire layer activation output during the forward pass of the classic Transformer [50] and BERT [11] model, respectively. Therefore, streamlining the activation memory requirement of the repeatedly-used softmax-based attention module is a simple yet effective way of improving the training memory efficiency of attention-based networks, e.g., the reduced peak-memory usage and memory footprint.

The experiment results demonstrate that the softmax output approximation drastically decreases the training memory required by the attention module by up to 84% while achieving competitive or improved performance of various attention-based models, i.e., basic Transformer [50], XLNet [52], and ALBERT [30], e.g., up to 5.4% higher accuracy, on machine translation [13], text classification [54], and sentiment analysis [34, 46] tasks. It validates that the softmax output approximation 1) can be readily applied to any existing attention-based models without modifying the network architecture, 2) does not significantly increase computation time for activation memory saving, and 3) enables models to learn longer sequences with a larger batch size when given the same memory budget. We implement the proposed softmax output approximation with PyTorch and make it public[1].

---

[1] https://github.com/eai-lab/SoftmaxOutputApproximation

## 2 Training of Softmax Operation

This section discusses the training procedure of the softmax operation and its memory usage.

**Softmax.** Let $\vec{z} = [z_1, z_2, \ldots, z_n]$ be a vector in $\mathbb{R}^{1 \times n}$, where $n$ denotes the sequence length. The vector $\vec{z}$ is obtained from the rows of $Z \in \mathbb{R}^{n \times n}$, which is the result of the query/key multiplication ($Z = QK^T / \sqrt{d_k}$). Then, the element $s_i$ in the softmax vector $\vec{s} = [s_1, s_2, \ldots, s_n]$ is defined as [4].

$$s_i = \frac{e^{z_i}}{\sum_{j=1}^n e^{z_j}} \text{ for all } i = 1, 2, .., n \text{ s.t. } 0 \leq s_i \leq 1 \text{ and } \sum_{i=1}^n s_i = 1 \tag{1}$$

Given the definition of softmax in Equation 1, let us assume there is a $l$-layer DNN (deep neural network) having a softmax operation at the $k$-th layer ($k < l$). To train the network via the gradient descent [42], the gradient of the loss $L(\vec{x})$ is computed with the input $\vec{x}$ w.r.t. the weight matrix $W_m$, i.e., $\partial L(\vec{x}) / \partial W_m$, at the $m$-th layer of the network ($m < k$). By using the chain rule, we get:

$$\frac{\partial L(\vec{x})}{\partial W_m} = \frac{\partial L(\vec{x})}{\partial \vec{z}_l} \frac{\partial \vec{z}_l}{\partial \vec{z}_{l-1}} \frac{\partial \vec{z}_{l-1}}{\partial \vec{z}_{l-2}} \cdots \frac{\partial \vec{s}}{\partial \vec{z}_{k-1}} \cdots \frac{\partial \vec{z}_m}{\partial W_m} \tag{2}$$

where $\vec{z}_l$ is the output of the $l$-th layer, $\vec{z}_k = \vec{s}$ and $\vec{z}_{k-1}$ in the middle is the output and input of the softmax operation at the $k$-th layer, respectively. Thus, to get $\partial L(\vec{x}) / \partial W_m$, the gradient of $\vec{s}$ w.r.t. $\vec{z}_{k-1}$, i.e., $\partial \vec{s} / \partial \vec{z}_{k-1}$, should be computed.

**Gradient Matrix.** To obtain $\partial \vec{s} / \partial \vec{z}_{k-1}$ in Equation 2, the partial derivative of the $i$-th softmax element $s_i$ w.r.t. its $j$-th input element $z_{k-1,j}$ is computed as follows:

$$\frac{\partial s_i}{\partial z_{k-1,j}} = s_i \left( \mathbb{1}_{i=j} - s_j \right) \tag{3}$$

where $\mathbb{1}_{i=j}$ is the indicator function. Since it only involves $s_i$ and $s_j$, not $z_{k-1,j}$, the gradient matrix of the softmax vector $\vec{s}$ w.r.t. its input vector $\vec{z}_{k-1}$ is given by:

$$\frac{\partial \vec{s}}{\partial \vec{z}_{k-1}} = \begin{bmatrix} s_1(1-s_1) & -s_1 s_2 & \cdots & -s_1 s_n \\ -s_2 s_1 & s_2(1-s_2) & \cdots & -s_2 s_n \\ \vdots & \vdots & \ddots & \vdots \\ -s_n s_1 & -s_n s_2 & \cdots & s_n(1-s_n) \end{bmatrix} \tag{4}$$

which allows the back-propagation of $\partial L(\vec{x}) / \partial W_m$ in Equation 2 if $\partial \vec{z}_l / \partial \vec{z}_{l-1}$ for all $l$ is available.

**Memory Requirement.** Thus, to compute $\partial \vec{s} / \partial \vec{z}_{k-1}$ in the back-propagation step, the softmax vector output $\vec{s}$ needs to be stored for each input $\vec{x}$ as an activation in memory during the forward pass of the network. Considering that 1) many DNNs, e.g., Transformers [50] and convolutional networks [26, 18], have deep and wide layers, and 2) back-propagation is usually performed with multiple examples (mini-batch), the total amount of activation memory required in training increases linearly to the number and length of softmax layers and the mini-batch size. For example, a network with $l$ softmax layers of length $n$ trained with a mini-batch consisting of $b$ examples requires a total of $l \cdot n \cdot b$ of activation memory for a single step of back-propagation.

**Memory Saving.** Motivated by this observation, we reduce the amount of activation memory required to back-propagate through a softmax layer by storing only a small subset of softmax output elements $\vec{s} = [s_1, s_2, ..., s_n]$ in memory, while evicting the rest of the elements from memory during the forward pass of the network. During back-propagation, the evicted elements are approximated such that the gradient matrix (Equation 4) obtained from the approximated softmax output minimizes the expected loss of training performance. The amount of activation memory saved by the softmax output approximation is multiplied by the number of softmax layers $l$ and the mini-batch size $b$, i.e., $l \cdot (n - m) \cdot b$, where $n$ is the total number of softmax output elements (the length of the attention score), and $m$ is the number of elements to be stored in memory. Also, the input of the softmax, $\vec{z}$, i.e., query/key multiplication $QK^T / \sqrt{d_k}$, which is the output of the previous layer, can be obtained from the approximated softmax output from the inverse of Equation 1 as:

$$z_i = \ln(s_i) + \ln\left(\sum_{j=1}^n e^{z_j}\right) \tag{5}$$

Thus, $\vec{z}$ of length $n$ can also be removed from the memory, which saves additional memory of $l \cdot n \cdot b$.

# 3 Softmax Output Approximation

## 3.1 Approximation Strategy

Given a softmax output vector $\vec{s} \in [0,1]^n$, its approximation, $\vec{s}'$, is computed with the following steps. First, among all softmax output elements of $\vec{s}$, the $m$ highest and $m$ lowest elements are stored in memory during the forward pass of the network. Then, the remaining $n - 2m$ elements are discarded from memory and approximated later in the backward pass. Lastly, using the stored and approximated elements together, the gradient matrix (Equation 4) is constructed for back-propagation (Equation 2).

**Approximated Softmax.** Given the softmax output vector $\vec{s}$, its approximation $\vec{s}'$ is defined as:

$$\vec{s}' = [\ \underbrace{h_1\ h_2\ ...\ h_m}_{m\ \text{highests}}\ \underbrace{s'_1\ s'_2\ ...\ s'_{n-2m}}_{n-2m\ \text{approximations}}\ \underbrace{\ell_m\ \ell_{m-1}\ ...\ \ell_1}_{m\ \text{lowests}}\ ] \tag{6}$$

where $h_1, h_2, ..., h_m$ and $\ell_m, \ell_{m-1}, ..., \ell_1$ are the $m$ highest and $m$ lowest elements, respectively, that are stored intact in memory, and $s'_1, s'_2, ..., s'_{n-2m}$ are the rest $n - 2m$ elements that approximate the non-stored elements, $s_1, s_2, ..., s_{n-2m}$. Figure 2 depicts an example of softmax output approximation.

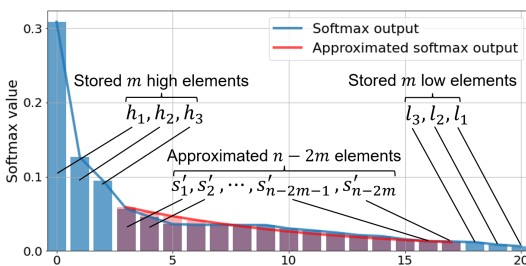

Figure 2: Among a total of $n$=21 softmax output elements sorted in descending order, the $m$=3 highest elements ($h_1, h_2, h_3$) and the $m$=3 lowest elements ($\ell_3, \ell_2, \ell_1$) are stored in memory. The remaining $n-2m$ elements ($s'_1, ..., s'_{n-2m}$) are not stored but approximated by the softmax output approximation function described below.

## 3.2 Element Selection for Memory Storing

Keeping the $m$ highest and $m$ lowest softmax output elements in memory minimizes the possible loss of training performance caused by the softmax output approximation. The following shows doing so minimizes the difference between the gradient matrix constructed from the original softmax $\vec{s}$ and the gradient matrix constructed from the approximated softmax $\vec{s}'$, so that the weight update computed from the approximated softmax becomes close to the original ones.

**Gradient Error.** Given a $n \times n$ softmax gradient matrix $\partial \vec{s}/\partial \vec{z}_{k-1}$ in Equation 4 and its approximate gradient matrix $\partial \vec{s}'/\partial \vec{z}_{k-1}$ obtained from the approximated softmax vector $\vec{s}'$, the gradient error between them is defined as $\epsilon(\vec{s}, \vec{s}') = |\frac{\partial \vec{s}}{\partial \vec{z}_{k-1}} - \frac{\partial \vec{s}'}{\partial \vec{z}_{k-1}}|$. We first decompose the gradient error $\epsilon(\vec{s}, \vec{s}')$ into each $i$-th row-wise error $\epsilon_i(\vec{s}, \vec{s}')$ consisting of 1) $i$-th column element's error and 2) the sum of the remaining $j \neq i$ column element's errors in the $i$-th row. Then, the row-wise error $\epsilon_i(\vec{s}, \vec{s}')$ for all $n$ rows are summed up to obtain the total gradient error, i.e., $\epsilon(\vec{s}, \vec{s}') = \sum_{i=1}^{n} \epsilon_i(\vec{s}, \vec{s}')$ as:

$$\epsilon(\vec{s}, \vec{s}') = \sum_{i=1}^{n} \epsilon_i(\vec{s}, \vec{s}') = \sum_{i=1}^{n} \left( \underbrace{|s_i(1-s_i) - s'_i(1-s'_i)|}_{i\text{-th element's error}} + \underbrace{\sum_{j=1 \backslash i}^{n} |s_i s_j - s'_i s'_j|}_{j \neq i\ \text{elements' error sum}} \right) \tag{7}$$

**Error Bound.** The second term $\sum_{j=1 \backslash i}^{n} |s_i s_j - s'_i s'_j|$ in Equation 7 is calculated in two cases. The set $S_1$ comprises indices where $s_i s_j - s'_i s'_j$ for $j \neq i$ is positive, while $S_2$ consists of the opposite.

$$\sum_{j=1 \backslash i}^{n} |s_i s_j - s'_i s'_j| = \begin{cases} |s_i(1-s_i) - s'_i(1-s'_i)| + 2\sum_{j \in S_1} s_i s_j - s'_i s'_j & \text{if } s_i(1-s_i) < s'_i(1-s'_i) \\ |s_i(1-s_i) - s'_i(1-s'_i)| - 2\sum_{j \in S_2} s_i s_j - s'_i s'_j & \text{if } s_i(1-s_i) \geq s'_i(1-s'_i) \end{cases} \tag{8}$$

If $s_i s_j - s'_i s'_j$ for all $j \neq i$ belongs to either $S_1$ or $S_2$, the second term of Equation 8 is bounded by its maximum value. In the case of $S_1$, it becomes:

$$2\sum_{j \in S_1} s_i s_j - s'_i s'_j = 2(s_i \sum_{j \in S_1} s_j - s'_i \sum_{j \in S_1} s'_j) = 2(s_i(1-s_i) - s'_i(1-s'_i)) \leq 0.5 \tag{9}$$

We denote the second term of Equation 8 as $\delta$. From Equations 7, 8, and 9, the gradient error $\epsilon(\vec{s}, \vec{s}')$ becomes to be bounded by:

$$\epsilon(\vec{s}, \vec{s}') = 2\sum_{i=1}^{n} |s_i(1-s_i) - s'_i(1-s'_i)| + \delta \leq 2\sum_{i=1}^{n} \max(s_i(1-s_i), 0.25 - s_i(1-s_i)) + 0.5 \tag{10}$$

where the inequality comes from the property of the softmax, i.e., $0 \leq s_i(1 - s_i) \leq 0.25$.

**Error Minimization.** Hence, to minimize the gradient error $\epsilon(\vec{s}, \vec{s}')$ in Equation 10, we select a set of $s_i$ as close as possible to some constants, i.e., $\{0, 1\}$ for the first term, $s_i(1 - s_i)$, and 0.5 for the second term, $0.25 - s_i(1 - s_i)$. First, by keeping the highest $m$ and the lowest $m$ elements in $\vec{s}$, we can select elements closest to the first two constants, $\{0, 1\}$. Second, similarly, when selecting the highest element $h_1$ from $\vec{s}$, it becomes the one closest to 0.5 among all softmax elements since the remaining $n-1$ elements are at most $1-h_1$ from the properties of softmax (Equation 1). Thus, their distances from 0.5 become at least $|h_1 - 0.5|$, i.e., $|0.5 - h_i| = |0.5 - (1 - h_1)| \geq |h_1 - 0.5|$ for all $i \neq 1$. It shows that selecting the $m$ highest and $m$ lowest softmax elements minimizes the gradient approximation error, $\epsilon(\vec{s}, \vec{s}')$, leading to the minimization of possible performance loss in training.

## 3.3 Approximation Method

During the back-propagation, the elements in the middle of Equation 6, i.e., $s'_1, s'_2, ..., s'_{n-2m}$, approximate the corresponding original elements, $s_1, s_2, ..., s_{n-2m}$, that are evicted from memory.

**Exponential Distribution.** We use the exponential distribution [43], i.e., $\lambda e^{-\lambda k}$, for the softmax approximation because 1) it naturally models a softmax in descending order well, 2) it can properly represent the non-linear changes of softmax as its slope changes rapidly; its derivative is $-\lambda^2 e^{-\lambda k}$, and 3) it can be efficiently computed with a small number of parameters. Unlike the original exponential distribution using a single parameter $\lambda$, we introduce two separate parameters $\alpha$ and $\beta$ to better represent the characteristics of the softmax operation in our approximation.

**Approximation Function.** Given the non-stored original softmax output elements sorted in descending order as $s_1 \geq s_2 \geq ... \geq s_{n-2m}$ that are computed but discarded from memory during the forward pass of the network, its $k$-th element $s_k$ is approximated as $s'_k$ in the backward pass as:

$$s'_k = \alpha e^{-\beta \cdot k} \text{ for } k = [0, 1, ..., N-1] \tag{11}$$

where $N = n - 2m + 2$ is the number of softmax output elements to be approximated, including $h_m$ and $\ell_m$ as the first and last reference values of the approximation, i.e., we let $s'_0 = h_m$ and $s'_{N-1} = \ell_m$. The two parameters $\alpha$ and $\beta$ are given by:

$$\alpha = h_m, \quad \beta = \frac{1}{N}\frac{1}{1-h_m}\left(\ln\frac{h_m}{\ell_m} + 2(h_m - h_{m+1})\right) \tag{12}$$

By using Equations 11 and 12, $s'_k$ for $k = 0, 1, ..., N-1$ can be computed to construct the gradient matrix in Equation 4 during back-propagation, while having only $2m$ values in activation memory, i.e., the $m-1$ highest and $m-1$ lowest softmax elements plus the two parameters $\alpha$ and $\beta$.

As mentioned above, the two parameters $\alpha$ and $\beta$ are defined such that the first and last approximation value becomes to be the same as $h_m$ and $\ell_m$, respectively, i.e., $s'_0 = h_m$ and $s'_{N-1} = \ell_m$. The parameter $\alpha$ can be easily obtained as $\alpha = h_m$ since $s'_0 = \alpha e^{-\beta \cdot 0} = h_m$. The parameter $\beta$ consists of three components, i.e., 1) normalization factor $\frac{1}{N}\frac{1}{1-h_m}$, 2) range value $\sigma$, and 3) rate value $\hat{\beta}$ as:

$$\beta = \frac{1}{N}\frac{1}{1-h_m} \cdot \sigma \cdot \hat{\beta} \text{ where } \sigma = -\frac{1}{\hat{\beta}}\ln\frac{1}{\alpha^2\hat{\beta}^2} \text{ and } \hat{\beta} = \sqrt{\frac{1}{h_m\ell_m}} \cdot e^{h_m - h_{m+1}} \tag{13}$$

**Range value ($\sigma$).** The range value $\sigma$ in Equation 13 is determined to map the domain of the approximation function, i.e., $k = [0, N-1]$, to a new domain $k' = [0, \sigma]$. Since the continuous domain of exponential distribution ranges from 0 to $\infty$, we define a new finite range of domain from which the corresponding approximations (codomain) are computed. To determine the range $\sigma$, we set the slope of the softmax approximation function in Equation 11 at $k = N-1$ as symmetrical to the slope at $k = 0$ to provide the length-invariant domain given an arbitrary length of softmax. Given the differentiation of $s'_k = \alpha e^{-\beta \cdot k}$ w.r.t. $k$ as $\partial s'_k/\partial k = -\alpha\beta e^{-\beta k}$, the slope at $k = 0$ becomes $\partial s'_0/\partial k = -\alpha\beta e^{-\beta \cdot 0} = -\alpha\beta$. By making the slope at $k = N-1$ as symmetrical to $-\alpha\beta$, i.e., $\partial s'_{N-1}/\partial k = -\alpha\beta e^{-\beta \cdot (N-1)} = -\frac{1}{\alpha\beta}$, replace $N-1$ with new domain $\sigma$, the range $\sigma$ becomes:

$$\sigma = -\frac{1}{\beta}\ln\frac{1}{\alpha^2\beta^2} \tag{14}$$

---

**Algorithm 1:** Model Training with Softmax Output Approximation

---

**Input:** $\mathcal{M}$: $l$-layer model, $\mathcal{B}$: training mini-batch, $m$: number of softmax elements to be stored in memory

**for** $k \leftarrow 1$ **to** $l$ **do**   [forward pass of the model $\mathcal{M}$]

   execute the operation at the $k$-th layer with $\mathcal{B}$;

   **if** *$k$-th layer == softmax* **then**

      **foreach** *training sample in $\mathcal{B}$* **do**

         ① store $m{-}1$ highest and $m{-}1$ lowest softmax output elements and $\alpha$, $\beta$ [Eq.12] in memory;

         discard the remaining $n{-}2m{+}2$ softmax output elements from memory;

         compute and store $\ln(\sum_{j=1}^{n} e^{z_j})$ [Eq.5] in memory;

      **end**

   **else**

      store the activation output of $k$-th layer obtained from $\mathcal{B}$ in memory;

   **end**

**end**

**for** $k \leftarrow l$ **to** $1$ **do**   [backward pass of the model $\mathcal{M}$]

   **if** *$k$-th layer == softmax* **then**

      **foreach** *training sample in $\mathcal{B}$* **do**

         ② approximate $n{-}2m{+}2$ softmax output elements using $\alpha$ and $\beta$ [Eq.11];

         make the softmax output vector $\vec{s}'$ from ①, ② [Eq.6];

         make the softmax input vector $\vec{z}$ from the softmax output vector $\vec{s}'$ [Eq.5];

         make the gradient matrix $\partial\vec{s}'/\partial\vec{z}_{k-1}$ from the softmax output vector $\vec{s}'$ [Eq.4];

      **end**

   **end**

   perform back-propagation for the $k$-th layer with its activation output;

**end**

---

**Normalization Factor.** As shown in Equation 13, the range $\sigma$ is normalized by $1/N$, as a total of $N$ softmax elements are approximated from it so that the discrete domain of $k = [0, 1, ..., N-1]$ can be used to approximate the $k$-th element. Since the approximation function in Equation 11 converges to 0 when $k \to \infty$, $\sigma$ is additionally multiplied with $1/(1-\alpha)$ where $\alpha = h_m$ (Equation 12) to increase the range $\sigma$ inverse proportional to $h_m$ based on the property that the sum of a softmax equals to one.

**Rate Value ($\hat{\beta}$).** The rate value $\hat{\beta}$ in Equation 13 determines the shape of the approximation function, playing a similar role to $\lambda$ in the exponential distribution. To determine $\hat{\beta}$, the approximation at $k = \sigma$ is set as the same as $\ell_m$ given Equation 11 and 14, i.e., we let $s'_\sigma = \ell_m$ and obtain $\beta$ as:

$$\beta = \sqrt{\frac{1}{h_m \ell_m}} \text{ from } s'_\sigma = \alpha e^{-\beta \cdot \left(-\frac{1}{\beta} \ln \frac{1}{\alpha^2 \beta^2}\right)} = \frac{1}{\alpha \beta^2} = \ell_m \tag{15}$$

where $\alpha = h_m$ as in Equation 12. Since the approximation is based on the exponential distribution, the convergence rate is proportional to $\beta$. To utilize this characteristic, the rate value $\hat{\beta}$ is multiplied by the distance between the $h_m$ and $h_{m+1}$ in the exponential function, which is given by:

$$\hat{\beta} = \sqrt{\frac{1}{h_m \ell_m}} \cdot e^{h_m - h_{m+1}} \tag{16}$$

By replacing $\beta$ with $\hat{\beta}$ in Equations 14 and 16 to distinguish them from $\beta$ on the left-hand side of the first equation in Equation 13, they become equivalent to the right two equations in Equation 13. Multiplying $\frac{1}{N}\frac{1}{1-h_m}$, $\sigma$, and $\hat{\beta}$ altogether in Equation 13 gives the final $\beta$ parameter in Equation 12.

**Approximation Overhead.** The proposed softmax output approximation 1) sorts the softmax elements, 2) performs the approximation, and 3) reorders the approximated softmax elements back to the original sequence, which takes $\mathcal{O}(n \log_2 n)$ of time complexity and $n \log_2 n$ bits of space (memory) complexity, respectively, where $n$ is the length of softmax vector. However, these overheads are negligible in general when compared to the massive amount of the training workload.

### 3.4   Model Training with Softmax Output Approximation

Algorithm 1 illustrates the model training procedure that uses the proposed softmax output approximation. Although its primary target is attention-based models (Transformers), it is applicable to any models having the softmax layers in them without modifying the network architecture. It reduces the amount of the softmax activation memory without affecting other parts of the network.

# 4 Experiment

**Implementation.** We implement a custom softmax operation that applies the proposed softmax approximation (Algorithm 1) with PyTorch [39]. Since our implementation is fully compatible with the existing normal softmax operation, it can be easily applied to any model that uses the original softmax by just replacing it with our custom softmax without changing any other model components.

**Tasks and Models.** As the attention mechanism is being widely used and developed primarily in the field of NLP, we evaluate the proposed softmax output approximation with various NLP tasks, i.e., machine translation (Multi30K [13]), text classification (AG News and DBpedia [54]), and sentiment analysis (SST-2 [29] and IMDb [34]) on different attention-based models, i.e., Transformer [50], XLNet [52], and ALBERT [30]. The details of each experiment are given below.

## 4.1 Activation Memory Saving and Model Performance

The first experiment is to evaluate the amount of activation memory reduction in the training process achieved by the proposed softmax output approximation and the consequent end-to-end model performance. To better understand its behavior, the models are 1) fine-tuned for down-stream tasks from a pre-trained model in Table 1, and 2) fully trained from scratch in Table 2 on an RTX A6000 GPU. We recompose the datasets of some tasks by limiting the maximum length of the input sequence due to the large capacity of attention-based models used in the experiment. The memory usage in the below three tables (Table 1(a), (b), and Table 2) represents the relative memory usage ratio to that of the normal softmax 1) without and 2) with the memory overhead incurred to save the softmax sequence, i.e., $n \log_2 n$, described above. We evaluate it with different sets of $m$ given a fixed $n$ in Equation 6 to restrict the activation memory usage of the softmax module during training and then measure their performance. The overhead remains constant with respect to $n$, regardless of $m$.

| Text Classification | | | | |
|---|---|---|---|---|
| Model | XLNet | | XLNet | |
| Task | AG News ($n$=100) | | DBpedia ($n$=100) | |
| | f1-micro | Mem usage (%) | f1-micro | Mem usage (%) |
| Normal | 0.9323 | 100% / 100% | 0.9889 | 100% / 100% |
| $m$=1 | 0.9175 | 1% / 11.9% | 0.9743 | 1% / 11.9% |
| $m$=5 | 0.9205 | 5% / 15.9% | 0.9869 | 5% / 15.9% |
| $m$=10 | 0.9335 | 10% / 20.9% | 0.987 | 10% / 20.9% |
| $m$=20 | 0.9353 | 20% / 30.9% | 0.9881 | 20% / 30.9% |
| $m$=30 | 0.9371 | 30% / 40.9% | 0.9877 | 30% / 40.9% |
| $m$=40 | 0.9226 | 40% / 50.9% | 0.9855 | 40% / 50.9% |
| $m$=50 | 0.9869 | 50% / 60.9% | 0.9862 | 50% / 60.9% |

| Sentiment Analysis | | | | |
|---|---|---|---|---|
| Model | ALBERT | | XLNet | |
| Task | SST-2 ($n$=100) | | IMDb ($n$=100) | |
| | f1-micro | Mem usage (%) | f1-micro | Mem usage (%) |
| Normal | 0.8326 | 100% / 100% | 0.8769 | 100% / 100% |
| $m$=1 | 0.8268 | 1% / 11.9% | 0.8833 | 1% / 11.9% |
| $m$=5 | 0.8314 | 5% / 15.9% | 0.8843 | 5% / 15.9% |
| $m$=10 | 0.8211 | 10% / 20.9% | 0.8826 | 10% / 20.9% |
| $m$=20 | 0.8257 | 20% / 30.9% | 0.8822 | 20% / 30.9% |
| $m$=30 | 0.8303 | 30% / 41.9% | 0.8840 | 30% / 41.9% |
| $m$=40 | 0.8360 | 40% / 50.9% | 0.8860 | 40% / 50.9% |
| $m$=50 | 0.8337 | 50% / 60.9% | 0.8838 | 50% / 60.9% |

(a) Text Classification result (fine-tuning)      (b) Sentiment Analysis result (fine-tuning)

Table 1: Among all $n$=100 softmax elements, only the $m$ highest and $m$ lowest elements are stored in memory. 'Normal' indicates the normal softmax that stores all output elements in memory.

**Text Classification.** Table 1(a) shows the relative activation memory usage ratio to the normal softmax operation and the f1-micro score [15] of two text classification tasks, i.e., AG News [54] and DBpedia [54]. They are fine-tuned from the pre-trained XLNet model [52]. In the AG News task, the 20.9% memory usage shows a slight difference (0.0012) in the f1-score from the normal softmax, i.e., 0.9335 vs. 0.9323. In addition, there is an improvement of 0.0546 when using 60.9% activation memory. The accuracy improvement can be regarded as one of the positive effects of the proposed approximation method as the approximation errors are broadcasted to the gradient matrix and act as white noise that can improve the learning performance. Similarly, in the DBpedia task, the 15.9% memory usage shows a slight difference (0.002) in the f1-score from the normal softmax, i.e., 0.9869 vs. 0.9889. It shows that sufficient training is achieved from a large number of approximated softmax elements and a small number of memory-stored elements (less than 5%) in text classification tasks. When the length of input sequence, $n$, is increased to 1000, the softmax activation output, which takes 34.33 GB of memory, is reduced to 8.80 GB (AG News task) and 7.08 GB (DBpedia task).

**Sentiment Analysis.** Table 1(b) shows the relative activation memory usage ratio to the normal softmax operation and the f1-micro score [15] of two sentiment analysis tasks, i.e., SST-2 [46] and IMDb [34]. They are fine-tuned from a pre-trained ALBERT [30] and XLNet [52] model, respectively. In the SST-2 task, the 11.9% memory usage shows a slight difference (0.0058) in the f1-score from the normal softmax, i.e., 0.8268 vs. 0.8326. When $n$ becomes 1000, the softmax activation output requiring 34.33 GB of memory space is reduced to 5.70 GB. In the IMDb task, the 11.9% memory usage shows a slight difference (0.0064) in the f1-score from the normal softmax, i.e., 0.8833 vs. 0.8769. The results of applying the softmax approximation show similar or higher scores to the normal softmax. They result from a combination of the characteristics of IMDb data, which contains many unstructured sentences and the noise caused by the softmax approximation.

| Machine Translation | | | | |
|---|---|---|---|---|
| **Model** | Transformer-base | | | |
| **Task** | Multi30k De-En ($n=40$) | | Multi30k En-De ($n=40$) | |
| | BLEU | Mem usage (%) | BLEU | Mem usage (%) |
| Normal | 36.52 | 100% / 100% | 34.05 | 100% / 100% |
| $m=1$ | 35.53 | 2.5% / 11.9% | 32.76 | 2.5% / 11.9% |
| $m=2$ | 36.29 | 5% / 14.4% | 33.97 | 5% / 14.4% |
| $m=4$ | 36.72 | 10% / 19.4% | 34.16 | 10% / 19.4% |
| $m=8$ | 36.46 | 20% / 29.4% | 34.28 | 20% / 29.4% |
| $m=12$ | 36.74 | 30% / 39.4% | 33.93 | 30% / 39.4% |
| $m=16$ | 36.41 | 40% / 49.4% | 33.96 | 40% / 49.4% |
| $m=20$ | 36.97 | 50% / 59.4% | 33.95 | 50% / 59.4% |

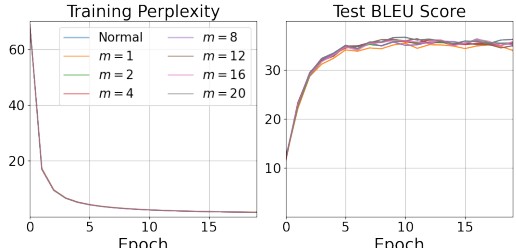

Table 2: Machine translation result (full training). Among all $n=40$ softmax elements, only the $m$ highest and $m$ lowest elements are stored in memory. 'Normal' indicates the normal softmax operation that stores all softmax output elements in memory for training.

Figure 3: The train perplexity (left) and test BLEU score (right) in machine translation (Table 2) over training epochs. The curves of the normal and approximated softmax almost overlap each other.

**Machine Translation.** Table 2 shows the relative activation memory usage ratio and the BLEU score [37] of two machine translation tasks, i.e., Multi30K on German-English (De-En) and English-German (En-De) translations [13]. Transformer basic models [50] are fully trained from scratch for them. As observed in Table 2, the 14.4% memory usage case shows only a small difference (0.23) in the BLEU score from the normal softmax, i.e., 36.29 vs. 36.52. When more than 19.4% of memory is allowed, it starts to outperform the normal softmax. The result demonstrates that the approximated softmax outputs enable efficient yet effective learning for full training of the Transformer model while requiring much less activation memory, which can be readily extended to other types of attention-based models in various architectures and sizes. Figure 3 plots their train perplexity and test BLEU score over training epochs, which shows almost the same learning trajectory is achieved when compared to the normal softmax for different $m$.

## 4.2 Softmax Output Approximation

The second experiment is to evaluate the approximation performance of the proposed method by comparing it against the original non-approximated softmax. We analyze the output of the softmax function over the training epochs. Figure 4(a) shows the comparison between the approximated softmax (Equation 6) and the normal softmax (Equation 1) at the 1, 8, 15, and 20 epochs. As shown in the figure, the approximation function effectively emulates the original softmax with low errors, i.e., only 0.0500 in MAE (mean absolute error) on average, and tends to get more correctly fitted as the training progresses.

## 4.3 Gradient Matrix Reconstruction

The third experiment is to evaluate the approximated gradient matrix (Equation 4) obtained from the approximated softmax output (Equation 6) by comparing it against the gradient matrix constructed from the original non-approximated softmax (Equation 1). Figure 4(b) depicts the heatmaps representing the gradient error (Equation 7) measured in MAE between them over the training epochs, showing the approximated gradient matrices are almost the same as the original non-approximated ones with consistently low errors. At the beginning of training (epoch 1), the average error is 0.00031, which is about 0.124% of 0.25, the maximum value that each gradient element can have. As the training progresses, the average error reduces to 0.088% (epoch 15), and most gradient elements in the remaining epochs retain low errors around 0.114%. Considering that the value of the gradient matrix is crucial as it is used in the back-propagation chain to compute the weight parameter update, it implies that the approximated softmax outputs enable the accurate update of the model weight parameters in the training process as it effectively reconstructs the gradient matrix with low error. Additionally, we convert the absolute gradient matrix error to the relative percentage error, i.e., 7.70% in epoch 1, 13.33% in epoch 8, 8.28% in epoch 15, and 12.74% in epoch 20. The elements of the gradient matrix range from -0.25 to 0.25, and since it has elements close to 0, the error appears to be unstable compared to MAE.

# 5 Related Work

The proposed softmax output approximation takes a unique approach orthogonal to many existing memory-efficient training techniques summarized below, enabling additional improvement in training memory efficiency when used on top of them, as they do not curtail the softmax activation memory.

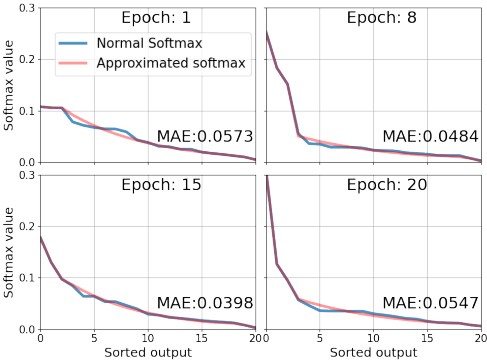
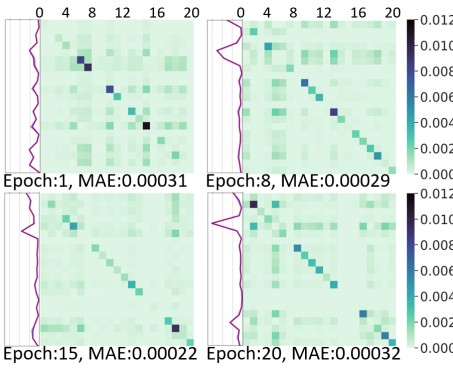

(a) Comparison of original and approximation          (b) The heatmaps of the gradient error (MAE)

Figure 4: (a) An example of the normal softmax (blue lines) and the approximated softmax output (red lines) sorted in descending order over the training epochs when $n$=21 and $m$=3. Among all 21 softmax elements, only 6 elements are stored in memory, while the remaining 15 elements are approximated with low error. (b) The heatmaps of the gradient error (RE) between the gradient matrix reconstructed from the approximated softmax outputs and the gradient matrix generated from the original softmax outputs. The attached plots on the left of each heatmap present the unsorted softmax output and the corresponding approximated softmax output, showing significant overlap.

**Diverse Approaches to Efficient Attention Mechanisms.** Efficiency improvements in attention-based models involve various approaches to streamline the attention mechanism, categorized as follows: 1) Efficient changes in the matrix-wise calculation of the attention mechanism, e.g., Flash Attention [9], Performer [8], Linear Attention [23], Random Feature Attention [40], etc. 2) By compressing the sequence to which the attention mechanism is applied, the size of the attention matrix is reduced, e.g., Longformer [2], Sparse Transformers [7], Reformer [24], etc. 3) Reducing the number of tokens in the attention mechanism, e.g., Linformer [51], Luna [33], Perceiver [19], etc. 4) Utilizing quantization for memory-efficient training, e.g., Mesa [36], GACT [32], etc.

Flash attention [9], an efficient matrix-wise calculation method, enhances the efficiency of the attention mechanism by employing varying memory access rates for each GPU memory layer. It speeds up memory-bound operations like matmul and softmax, where memory access exceeds computational requirements. Tiling enhances the efficiency of matrix matmul, whereas softmax is optimized through recomputation. To save memory and avoid the $O(n^2)$ memory requirement for softmax recalculation, the input is reconstructed after storing softmax normalization statistics. In this method, the proposed softmax approximation has limited applicability due to the absence of storage or processing of softmax output.

In Performer [8], an efficient matrix-wise calculation method, the row vectors of the query and key are mapped from $d$ dimensions to $r$ dimensions using a random feature map $\phi$. Then, attention is computed by taking the inner product of the mapped $\phi(q_i)$ and $\phi(q_j)$. The memory efficiency of this approach differs from the proposed softmax approximation method since it eliminates the softmax step in the process of streamlining matrix-wise operations.

Longformer [2], a sequence compression method, utilizes techniques like dilated sliding and global attention to overcome the limitations of existing methods, which face challenges in learning lengthy sentences due to their $O(n^2)$ complexity. This approach proposes compressing the attention matrix from a larger size to a smaller one, effectively resolving the issue. In this approach, the operation of $QK^T$ is streamlined, and the attention score is computed using softmax in the same way as existing attention mechanisms. Based on this, the proposed softmax approximation method can be utilized in this process. It efficiently approximates the whole from a part of the softmax output, making it suitable for restoring an already compressed sequence matrix, regardless of the input type.

Linformer [51], a method for reducing the number of tokens, simplifies complexity by decomposing the attention matrix into low-rank matrices and performing operations on these low-rank matrices. The results obtained from these low-rank matrix operations are then processed through a softmax operation to determine the attention scores matrix. This method overlaps with the proposed softmax approximation method due to the presence of activation memory resulting from the softmax operation.

Mesa [36], a memory-saving method that employs quantization, utilizes quantization to store low-precision activation outputs, thereby reducing the memory used for activation storage during forward propagation. Additionally, during back-propagation, these low-precision activation outputs are dequantized to compute the gradients. In the context of memory-saving through quantization, it's evident that efficiency limitations depend on the physical memory unit. However, differences arise in the proposed softmax approximation method because it can adapt flexibly based on the sentence.

The various methods described above have been employed to enhance efficiency through hardware-dependent measures, matrix-wise calculation efficiency, and the compression or removal of segments within the attention matrix. The softmax approximation proposed in this paper is orthogonal to such approaches, as it improves activation memory efficiency by approximating the entire softmax output from a subset. Notably, it can be used without requiring structural modifications to the existing model or attention mechanism. These distinctions make it easily applicable to many widely used attention-based models, and it can also be used concurrently with the above-mentioned efficient methods.

**Re-materialization.** Instead of saving activation outputs of each layer in memory, re-materialization methods [6, 10], also referred to as gradient checkpointing [16, 14, 1, 28], generate them on the fly by re-executing (re-computing) the network. Although a substantial amount of memory can be saved by re-execution of the network, it comes at the cost of increased computation time (i.e. time-memory trade-off) as the network is repeatedly re-executed proportional to the number of model layers. Since 1) the complexity of the attention module is high, i.e., $\mathcal{O}(n^2)$ in general [50], and 2) it appears multiple times in the network architecture, re-materializing attention-based networks is not feasible in practice. Also, to handle the time-memory trade-off, an appropriate schedule should be chosen for a back-propagation graph [27, 17]. Unlike them, the proposed softmax output approximation 1) saves the activation memory in training without increasing computation as the network is executed only once, not multiple times, and 2) does not require scheduling a back-propagation graph.

## 6 Discussions and Limitations

**Approximation Overhead.** To implement the proposed softmax approximation using the exponential distribution, which is a decreasing function, the softmax output elements are initially sorted in descending order. After approximating the sorted softmax output, they are reordered back to their original sequence to perform back-propagation. In terms of space (memory) complexity, it requires storing the original sequence of softmax elements in memory, taking $n \log_2 n$ bits, where $n$ is the number of softmax elements. When it comes to time (computation) complexity, it involves 1) sorting the softmax elements, 2) performing the approximation, and 3) reordering the approximated softmax elements back to the original sequence.

The optimal implementation of the softmax approximation requires a lower-level programming language, like the CUDA kernel, rather than Python. Thus, the above processes would better be carried out at the 'CUDA kernel' level to properly implement the softmax output approximation and assess the actual overhead. In this paper, we implement the softmax approximation in a non-optimal way using Python, unlike the original softmax function highly optimized with the CUDA kernel. Our ongoing work involves further refinements, and one of the objectives is to optimize the softmax approximation algorithm with CUDA kernels. That said, the aforementioned overheads are negligible in general when compared to the massive training workload. Hence, with relatively little overheads in both memory and computation, the proposed method can efficiently approximate the softmax output.

**Performance Improvement.** In some cases, we have observed that the model's performance improves when the proposed softmax approximation is applied. When certain approximation errors in the softmax output are confined to a small range, they behave like white noise when delivered and broadcast to the gradient matrix. This white noise effect can have a positive impact on performance.

## 7 Conclusion

We propose to approximate the softmax output in the attention module to reduce its activation memory required to train attention-based deep models. By storing only a subset of the softmax output elements in memory during the forward pass and approximating the evicted elements during the backward pass, a significant amount of activation memory can be saved in an architecture-agnostic way when training attention-based deep models. The experiment shows that the proposed softmax output approximation, which significantly reduces softmax activation memory by up to 84% while maintaining comparable performance, is likely to be effective for models dealing with large datasets and substantial memory.

## Acknowledgments and Disclosure of Funding

This work was supported by the National Research Foundation of Korea(NRF) grant funded by the Korea government(MSIT) (RS-2023-00277383) and Institute of Information & communications Technology Planning & Evaluation(IITP) grant funded by the Korea government(MSIT) (No.2020-0-01336, Artificial Intelligence graduate school support(UNIST)).

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
