# OpenReview forum: "Softmax Output Approximation for Activation Memory-Efficient Training of Attention-based Networks"
_NeurIPS.cc/2023/Conference — NeurIPS 2023 poster_

### Official Review · Reviewer_8HU2 · 2023-07-04

**Soundness:** 3 good
**Presentation:** 3 good
**Contribution:** 3 good
**Rating:** 7
**Confidence:** 5

**Summary:**


This paper proposes the Softmax Output Approximation algorithm, which approximates the softmax output during the forward pass and reconstructs it during the backward pass. In attention-based models like Transformer, the softmax output activations consume a significant amount of memory. By approximating them, a considerable amount of training memory can be saved. The algorithm successfully achieves memory savings of up to 84% compared to the existing softmax in various tasks.

**Strengths:**

1. The paper is well-written and easy to understand overall.

2. The mathematical analysis of gradients for approximating softmax outputs is reasonable. In particular, **the highly novel aspect of this approach** is that it approximates the distribution by assuming it and then uses sorting, without relying on pruning or quantization techniques.

3. The experimental results demonstrate that the proposed method effectively reduces memory usage during pretraining or fine-tuning on certain datasets while achieving comparable accuracy to the baseline.





**Weaknesses:**

**1. Low accuracy on IMDb tasks**

: The paper acknowledges the low performance on certain datasets, such as IMDb, and attributes it to the specific characteristics of the task. However, if the proposed softmax approximation struggles with tasks like IMDb, it may be challenging to apply it to other difficult tasks.

**2. Approximation time overhead**

: The paper claims that the overhead of the approximation is negligible compared to the general training workload. However, considering the potential overhead caused by sorting when dealing with large softmax outputs, it raises doubts about whether the overhead can truly be disregarded.

**3. Lack of Comparison**

: There are existing algorithms that approximate activations in various ways. For example, ActNN [1] and GACT [2] compress all activations based on their sensitivity, while [3] Mesa compresses all activations in a Transformer-specific manner. AAL [4] and DropIT [5] approximate input activations of linear and convolutional layers using auxiliary activations and pruning, respectively. It would be beneficial to compare the proposed approach with these existing methods.

[1] ActNN: Reducing Training Memory Footprint via 2-Bit Activation Compressed Training

[2] GACT: Activation Compressed Training for Generic Network Architectures

[3] Mesa: A Memory-saving Training Framework for Transformers

[4] Learning with Auxiliary Activation for Memory-Efficient Training

[5] DropIT: Dropping Intermediate Tensors for Memory-Efficient DNN Training

**Questions:**

I think the idea presented in this paper is highly novel. If the following questions are addressed satisfactorily, **I am willing to raise my score to 5-7**.

**1. Low accuracy on IMDb tasks**

:It is disappointing to see a significant performance decrease on the IMDb dataset, while the SST-2 dataset shows minimal performance degradation. In my opinion, with appropriate hyperparameter tuning, it should be possible to achieve performance comparable to the baseline. If it is not feasible, it would be helpful to provide a qualitative explanation that softmax output activation may work well for general tasks, and IMDb tasks are an exceptional case.

**2. Approximation time overhead**

:It would be beneficial to report the increase in training time compared to the baseline. Specifically, it would be valuable to demonstrate that the approximation time overhead remains low even for larger input sizes.

**3. Lack of Comparison:**

Adding various algorithms [1-5] that approximate activations to the Related Work section would be beneficial. Particularly, for GACT [2] and Mesa [3], which also target Transformer and approximate softmax outputs, it is important to provide specific comparisons with proposed Softmax Output Approximation algorithm in terms of accuracy, memory usage, and training time.

**Limitations:**

This paper lacks a discussion on the limitations of the proposed algorithm and possible future work. It would be beneficial to include these aspects in the revised version of the paper.

---

> ### Author Rebuttal · Authors · 2023-08-08
>
> We deeply appreciate the constructive feedback and the clear guidelines provided by the reviewer to make our work better.
>
> **Q1** Low accuracy on IMDb tasks
> > We have conducted a new experiment with the IMDb task and obtained comparable performance to the baseline, i.e., baseline accuracy 0.876 vs. our method accuracy 0.883 ($m=1$), by tuning the hyperparameters as the reviewer suggested. Please take a look at the details of the new experimental result (PDF file) posted on the Global rebuttal. The reason we got the low-performance result on the IMDb task in our submission might be some subtleties and difficulties in approximating the exact softmax outputs that tend to diverge randomly from the (continuous) exponential function. The IMDb task was the one that revealed this phenomenon most in our experiment. Although our method now works well with the IMDb task, as shown in our new experiment, we will keep working on validating the efficacy of our method for other difficult and general tasks and include them in our revised paper.
>
> **Q2** Approximation time overhead
> > As the reviewer correctly pointed out, our method inevitably increases the training time to some extent because it includes additional sorting and softmax output approximation processes. As the reviewer suggested, we have measured the time overhead incurred by our method over different batch sizes, sequence (input data) lengths, and the number of softmax output values stored in memory, of which details are included in the PDF file in the Global rebuttal. From this measurement experiment, we can observe that 1) it takes almost the same amount of time no matter how many softmax output values are stored in memory given the same input data length, and 2) the increase in time complexity caused by the change in batch size is greater than the sequence length. However, it looks like that our method takes at least about 1.6$\times$ and up to about 6$\times$ more time when compared to the baseline that does not apply our methods. The reason for it, we speculate, is that our method is implemented with the relatively slow and naïve “Python (Pytorch)” rather than fast “CUDA kernel (C++)”, unlike the original softmax operation that is highly optimized and run with “CDUA kernel”. In more detail, our method simply selects the “to-be stored softmax output values” after sorting them in the forward pass. And then, during the backward pass, it approximates the non-stored softmax output values and places them into their original positions. When turning this algorithm into actual code, we needed to use many additional matrix operations such as slicing, the scatter function, etc., not only the sorting mechanism, which takes a much longer time than sorting several hundreds or thousands of softmax elements, without implementation optimization (e.g., CUDA kernel optimization). That might explain the increased time complexity of our method compared to the baseline. Since the sorting time complexity we used is just $n \log_{2} n$, which does not seem to incur significant time overhead compared to the compute-intensive back-propagation (gradient computation) when $n$ is around several thousands, we believe that the time gap between our method and the baseline would be dramatically reduced through the implementation optimization of our method, making our method more practical. Thanks again for your careful comment on this matter, and we will keep working on how to optimize our method to decrease the time overhead.
>
> **Q3** Lack of Comparison
> > We appreciate the reviewer providing reference papers that can be of great help to our submission and future research. We will include those papers in our related work and thoroughly discuss them against our method. Also, we will conduct a comparison experiment in terms of accuracy, memory usage, and training time. Here, we briefly discuss them as follows.
>
> > In the case of GACT, to our understanding, unit bits are reduced through quantization with a maximum of 4 bits and a minimum of 2 bits to minimize overlapping and errors of each layer under specific conditions. Thus, when applied to the softmax layer, it could achieve 16x memory reduction at maximum, i.e., from 32 bits (floating point) to 2 bits. And in the case of Mesa, similar to GACT, the gradient is quantized. As a preliminary experiment, we have briefly checked the difference between the gradient matrix obtained from the softmax output approximated by our method and the gradient quantized from Mesa. Although additional experiments seem necessary as it is unclear how this will affect actual learning, their gradient values generated from the same input data do not differ significantly from the result of our preliminary experiment. In other words, the error rates of the gradient matrix created from the existing normal softmax and the gradient matrix from Mesa and our method seem to be similar, meaning that there may be a little difference in their model accuracy.
>
> > In sum, we speculate our method would have more room to reduce softmax activation memory thanks to its flexible ratio of approximation ($m$ can be changed accordingly given $n$) while achieving similar model accuracy to GACT and Mesa, as the gradient matrix obtained from our method and Mesa seem to have similar error from the gradient obtained from the original softmax operation.
> However, we will conduct in-depth experiments with GACT and Mesa and compare them against our method (including training time) to see if our conjecture is correct or not, which will be included in our revised paper.

---

> > ### Comment · Reviewer_8HU2 · 2023-08-17
> > **Response to Reviewer**
> >
> > I sincerely appreciate the author's effort and response.
> >
> > All my queries have been perfectly resolved. I commend you sincerely for tackling what must have been a very difficult process of tuning the hyperparameters in the IMDb dataset to obtain proper accuracy, as per my request.
> >
> > I believe **the algorithm's approximation method is a highly novel approach that has not been proposed before**, and **it maintains high accuracy while saving memory**. I strongly recommend to the AC that this paper be accepted, and I will raise the score to 7.

---

> > > ### Author Response · Authors · 2023-08-18
> > >
> > > We really appreciate your acknowledging the novelty of our work and your willingness to recommend it to be accepted by NeurIPS 2023. Your support gives us stronger confidence in our work.
> > >
> > > We are grateful for your suggestion to tune the hyperparameters for the IMDb task, which helps us gain more confidence that our method would be applied to more difficult and general tasks. We will conduct additional experiments on this and include the results in our revised paper.
> > >
> > > Also, we will compare our work against GACT and Mesa, which, we expect, will effectively distinguish our method from the existing works on approximating activations.

---

### Official Review · Reviewer_r1xy · 2023-07-06

**Soundness:** 4 excellent
**Presentation:** 4 excellent
**Contribution:** 4 excellent
**Rating:** 8
**Confidence:** 3

**Summary:**

This paper tries to reduce the memory footprint while training Transformer models by only keeping a fraction of softmax output and estimate them back during back propagation.

**Strengths:**

This is a very interesting and practical topic. With this tech, we can reduce the computation cost significantly while training large models.

**Weaknesses:**

I don't see any obvious weakness.

**Questions:**

Could you report the training profiling comparisons w/ and w/o the approximation overhead?

Could you elaborate more on how this tech works with other Transformer variants like Performer?

---

> ### Author Rebuttal · Authors · 2023-08-09
>
> We wanted to take a moment to express our heartfelt gratitude for reviewing our paper. Received your positive assessment has given us renewed confidence in our work.
>
> **Q1** Training profiling
> > Thanks for a good question. We felt that the training profiling the reviewer mentioned means two aspects (model performance and training time). Please let us know if we have mistaken your question.
>
> > First, profiling in terms of model performance (accuracy) is expected to be acceptable through our experiments presented in our current manuscript. In addition, in the case of the IMDb task, which had a rather low score, we have re-experiment it, which shows comparable performance to the baselines. Our Global rebuttal response includes the details of the new experiment on the IMDb task.
>
> >Second, profiling in terms of time is deemed necessary. Basically, we expect that the time complexity increases when applying our method, compared to the baseline that does not apply our method, as our method needs to perform additional processes such as sorting. Regarding this, we have conducted a new experiment on time overhead and included the result in the Global rebuttal response. We kindly ask the reviewer to check out the Global rebuttal response.
>
> **Q2** Transformer variants
> > Please let us describe the gist of common Transformer variants and their applicability to our method. The biggest goal of the Transformer variants is to reduce the time complexity of $O(n^2)$ of the matrix-wise calculation ($QK^T$) of the attention mechanism in the inference phase. In addition, some of these methods of reducing time complexity also streamline all of the $softmax(\frac{QK^T}{\sqrt{d}})V$, which are the computational processes of the attention mechanism. However, since these attention mechanisms are new variants of attention mechanisms, they are often accompanied by particular models suitable for them. However, our method can be applied in the case of an efficiency method with no change to softmax among attention mechanisms. The reason is that the applicability of our method depends on the existence of a softmax function for the input. If the softmax function exists and takes up activation memory, we can try our method. Also, the advantage of our method is that it can be applied to all variants of attention-based models. We believe this has been proven by our experiments on XLNet (0.12 billion parameters) with a similar number of parameters to GPT-2 small (0.124 billion parameters).

---

> > ### Comment · Reviewer_r1xy · 2023-08-18
> >
> > Thanks authors for the response! My questions have been addressed.

---

> > > ### Author Response · Authors · 2023-08-19
> > >
> > > We are glad that our answer has helped address the reviewer’s concerns and questions.
> > >
> > > We will include how our method can be applied to other Transformer variants and conduct experiments on it. We appreciate the reviewer’s positive feedback and support for our work.

---

### Official Review · Reviewer_s2ms · 2023-07-06

**Soundness:** 4 excellent
**Presentation:** 4 excellent
**Contribution:** 3 good
**Rating:** 7
**Confidence:** 4

**Summary:**

The paper proposes an effective scheme for compressing the backpropagation of Softmax, the main idea of which is to only retain the maximum and minimum m values of the output, with the middle part using linear interpolation. Impressively, this approximation scheme not only saves memory but also achieves better results than precise backpropagation in some tasks.

**Strengths:**

1. The memory usage of Softmax backpropagation is effectively reduced through approximation, which can be used in Attention models;

2. There is a strict theoretical analysis of approximation errors;

3. The performance is even better than the accurate Softmax backpropagation in some tasks.

**Weaknesses:**

1. There is no quantitative analysis of the amount of memory that can be saved theoretically;

2. The memory efficiency has not been validated on larger models;

3. The reason why the approximate softmax is better (in some tasks) has not been further analyzed.

**Questions:**

The method in the paper does indeed save the memory required for the backpropagation of Softmax, but I have a question, that is, is the proportion of memory occupied by the Softmax part in the entire backpropagation crucial?

Intuitively, the memory occupied by the Softmax part is not too much, but perhaps my intuition is wrong, so I think this part needs further quantitative estimation, which determines the importance of this work (especially in the current large language model scenario).

**Limitations:**

I couldn't find any discussion about limitations in the main text and the appendix of the paper. Perhaps the authors should include it in the paper.

---

> ### Author Rebuttal · Authors · 2023-08-08
>
> We thank the reviewer for providing valuable and positive comments regarding our work, and the questions that helped us to refine our work. Here, we have provided a detailed response.
>
> **Q1** The proportion of memory occupied by the softmax
> > Thanks for asking a crucial question about our work. It may seem the proportion of the softmax output is not significant from the perspective of the entire network. However, our analysis and experiment show that it actually takes a large portion of activation memory. For instance, the softmax operation takes up 80% of the attention module itself, and 64.7% and 62.2% of the entire layer activation output in the forward pass of the classic Transformer and BERT model. However, to validate the effectiveness of our method, we will conduct in-depth experiments on large models, as the reviewer suggested.
>
> > In principle, the size of the softmax activation memory of an attention-based model is equivalent to [Batch size $\times$ Multi-head $\times$ Sequence length $\times$ Sequence length]. Therefore, for large language models that take longer sequences and/or a larger number of multi-heads, the percentage, as well as the absolute size, of softmax activation memory can be easily increased.
>
> > Although we will conduct experiments on larger models, we can take a simple example for an intuitive understanding of how big is the softmax activation output. In the case of the XLNet model, the softmax operation is passed 12 times for each forward execution. By assuming that the batch size is 256, the multi-head is 12, and the sequence length is 100, the total amount of memory taken by the softmax output is [256 $\times$ 12 $\times$ 100 $\times$ 100] $\times$ 12 $\times$ 4 bytes, which is about 1.37GB. We can expect that for a large language model with more encoders, decoders, and multi-heads, softmax activation will take up more portion of activation memory.

---

### Official Review · Reviewer_UEEi · 2023-07-07

**Soundness:** 2 fair
**Presentation:** 2 fair
**Contribution:** 3 good
**Rating:** 4
**Confidence:** 4

**Summary:**

This paper proposes to approximate softmax to improve memory efficiency for training attention models. During forward pass, first the softmax output is computed. Only the $m$ highest and $m$ lowest elements are stored along with sorted order from the the $n$ elements. Rest of the $n-2m$ entries are discarded to reduce the memory footprint. During backward pass, the discarded entries are approximated using exponential distribution whose parameters can be approximated using the stored elements. The $m$ highest and $m$ lowest elements are selected so as to minimize error in approximated gradient during the backward pass. Experiments on machine translation,  text classification, and sentiment analysis showcase the efficacy of proposed softmax approximation.

**Strengths:**

- While the idea to approximate softmax using the top-$m$ entries to reduce attention memory footprint has been explored before, the use of $m$ lowest entries and its motivation from gradient error minimization is novel.
- The proposed method is simple and easy for the community to reproduce. The method also offers an easy way to plugin the approximation to pre-trained models.

**Weaknesses:**

  - Incorrect claims and/or missing justifications
    - Lines 286-290 Efficient attention models do not focus on training memory reduction. This statement is false as there are many efficient attention mechanisms that explicitly reduce the memory for attention by (a) changing the order of computation with kernelized softmax: Linear [1], Performer [2], RFA [3]; (b) reducing the number of tokens to attend: sparse [4], Reformer [5], local [6], longformer [7], or (c) condensing the sequence: Linformer [8], compressive [9], perceiver [10]
    - Line 60-61:  the softmax operation takes up 80% of the attention module itself, and 64.7% and 62.2% of the entire layer activation output during the forward pass. It is not clear that at what sequence lengths is this claimed.
    - Error bound (Line 133): The assumption $s_i s_j \geq s'_i s'_j$ or $s_i s_j \leq s'_i s'_j$ is missing justification or any discussion. I am not sure how well this is going to hold in practice.
  - Notations:
    - Sometimes the paper uses confusing notations. For instance see line 74, $z$ is defined to be a vector of length $n$ with query/key multiplication $z= QK^T$ as an example. Typically $Q$ and $K$ are matrices of size $n \times d$ which would lead $z$ to be a matrix.
- Weak experiments:
    - Baselines: The paper only compares against the scaled dot product attention as the baseline. The following two baselines are particularly close and also target memory reduction.
      - FLASH Attention [11] exploits lazy softmax computation (normalization can be delayed until the end of the attention) to avoid storing softmax values and directly compute output values leading to memory savings.
      - Memory-efficient Transformers via Top-k Attention [12]. This is another closely related work that not only targets memory improvements for attention but also improves the memory for feed-forward layers.
  - Tasks:
      - While the paper conducts experiments on MT, text classification, and sentiment analysis, it is unclear how the proposed technique would impact more general tasks in NLP or other domains. More real world tasks such as pre-training would help understand this better. Either Long range Arena [13] or more recent Comprehensive Attention Benchmark [14] would also provide more coverage on tasks and help understand better the tradeoffs of softmax approximation.
      - Time-memory trade off as a function of sequence length. While the paper presents the relative memory improvement for sequence length of $100$, it would be good to see a general trend for memory consumption and time computation overhead as sequence length varies. The overhead on computation time is not discussed anywhere in the paper.
- Missing references to other works that approximate softmax to improve the computational and memory complexity. Some of these work could also be good candidates as baselines
    - SMYRF: Efficient Attention using Asymmetric Clustering (NeurIPS 2020)
    - Fast transformers with clustered attention (NeurIPS 2020)
    - Nystromformer (AAAI-21)
    - Sparse attention with learning-to-hash (ICLR 2022)

*References*:
- [1] Fast Autoregressive Transformers with Linear Attention (ICML 2020)
- [2] Rethinking Attention with Performers (ICLR 2021)
- [3] Random Feature Attention (ICLR 2021)
- [4] Generating Long Sequences with Sparse Transformers (Arxiv 2019)
- [5] Reformer: The Efficient Transformer (ICLR 2020)
- [6] Generating wikipedia by summarizing long sequences (Arxiv 2018)
- [7] Longformer: The Long-Document Transformer (Arxiv 2020)
- [8] Linformer: Self-Attention with Linear Complexity (Arxiv 2020)
- [9] Luna: Linear Unified Nested Attention (NeurIPS 2021)
- [10] Perceiver: General Perception with Iterative Attention (ICML 2021)
- [11] Fast and Memory-Efficient Exact Attention with IO-Awareness (NeurIPS 2022)
- [12] Memory-efficient Transformers via Top-k Attention (Proceedings of the 2nd Workshop on Simple and Efficient Natural Language Processing 2021)
- [13] Long Range Arena : A Benchmark for Efficient Transformers (ICLR 2021)
- [14] CAB: Comprehensive Attention Benchmarking on Long Sequence Modeling (ICML 2023)

**Questions:**

Please see the discussion on weakness. In particular on missing justifications/claims, comparison to baselines/LRA tasks, and time-memory trade-off.

**Limitations:**

The discussion on negative societal impact is not required.

---

> ### Author Rebuttal · Authors · 2023-08-09
>
> ### Incorrect claims and/or missing justifications
> **W1** Line 286-290
> > Thanks for letting us know about the related works we missed. What we meant was that existing works on efficient Transformers do not explicitly try to save activation memory, unlike our method. Although they can save memory not in terms of activation memory, we would like to mention that our method is unique and orthogonal to them. By the way, we will go over them and revise our paper. In Performer[2] in (a), the row vector of query and key is mapped from $d$ dimension to $r$ dimension through random feature map $ϕ$. Then, attention is calculated by inner producing the mapped $ϕ(q_i)$ and $ϕ(k_j)$. In this case, memory efficiency is different from our method because the matrix-wise operation of the attention mechanism is removed from the efficient softmax. In Longformer[7] in (b), the existing attention-based model fails to learn long sentences due to the complexity of $O(n^2)$. This is a method using dilated sliding, global attention, etc. In this case, the operation of $QK^T$ is streamlined, and the attention score using softmax is obtained in the same way as the existing attention. We may be able to apply our method to this process. This will be tested later and used for comparison with transformer variants. In Linformer[8] in (c), since it simply decomposes the attention matrix into a low-rank matrix and uses it to obtain an attention score, it seems that it can be used overlapping with our method applicable depending on the presence or absence of softmax for attention score. Thank you for letting us know about the various perspectives of memory efficient attention as above. They may be different from the efficiency of activation memory that occurs during the training process like our method, but they will be greatly helpful for our future research.
>
> **W2** Line 60-61
> > It is the ratio of how much softmax is passed through among the layers that each input passes through inference in BERT and Transformer-base. It has nothing to do with the length of the sentence.
>
> **W3** Line 133 Error bound
> > As the reviewer correctly pointed out, the assumption may not always be held. However, we expect that the error bound does not change even if the assumption does not hold. That is because 1) the error bound is the sum of absolute error and 2) both the original softmax output ($s_i$) and the approximated output ($s'_i$) are positive numbers with a value between 0 and 1. Thus, the total amount of error given on the left-hand side in Equation 8 should not change regardless of the signs of each term. The reason we explicitly add the assumption is that we hope it would helps readers understand the process of the error-bound analysis. Although we believe the assumption does not affect the error bound, we will keep looking into it to see if there is a mistake or something we are missing. We appreciate again your careful and crucial comment on this.
>
> **W4** Line 74 Notations
> > Thank you for your accurate point. In the case of that notation, it's our mistake. Since $Z$ is a matrix and $\vec{z}$ is a vector, as you said  $Z=QK^T$ of $Z \in \mathbb{R}^{n \times n}$. Here, $n$ is the sequence length, and since softmax is applied based on $\vec{z_i} \in \mathbb{R}^{1\times n}$ to obtain the attention score, $\vec{z}$ is expressed as $\vec{z} \in \mathbb{R}^n$. Looks like it needs a fix.
>
> ### Weak experiments
> **W1** Baselines
> > Flash attention speeds up the attention mechanism by using different memory access rates for each memory layer on the GPU. To improve efficiency using different access speeds, it accelerates matmul and sofmtax, that is, memory-bound operations, in which memory access is greater than the amount of computation. For this purpose, tilling and recomputation were used. In the case of matrix matmul, it is efficiently changed using tilling, and in the case of softmax, it was made efficient through recomputation. In this case, in order to avoid using $O(n^2)$ memory required to recalculate the softmax, the input is reconstructed after saving the softmax normalization statistics. In this case, our method is less applicable as there is no storage or operation of the softmax output. If possible, we think that we can increase the storage efficiency by not storing the input and normalization statistics and applying our method.
>
> **W2** Tasks
> > Our experiments were evaluated against representative NLP tasks. In the case of Text Classification and Sentiment Analysis, the pre-training model was fine-tuned, and in the case of MT, full-training was performed, which implicitly includes the pre-training stage. As the reviewer pointed, we were unable to experiment with a wide range of tasks due to time and resource limitations. Nevertheless, our method can restore the original softmax outputs using only a part of the softmax output, so it is expected to be capable of providing a similar performance unless data modality or type changes significantly. We will continue to conduct verification experiments with various tasks, including many interesting works the reviewer introduced, to see if there is some deviation from our expectations and revise our manuscript accordingly.
> > For Memory-efficient Transformers via Top-k Attention[12], a sparse attention method similar to Longformer[7] is borrowed from "Generating Long Sequences with Sparse Transformers" for softmax efficiency. We will evaluate the applicability and comparison of our method. We will also check the benchmarks and methods presented in the paper as much as possible we can.
>
> **W3** Time-memory trade off
> > We have attached a PDF file in Global rebuttal response, which provides new experiment results on memory consumption and computation time. We kindly ask the reviewer to see the Global response.

---

> > ### Comment · Reviewer_UEEi · 2023-08-17
> > **Thank you for clarification**
> >
> > I would like to thank the authors for taking out time to address the questions. However, I still have unresolved questions and concerns.
> >
> > **Lines 286-290**
> > >  What we meant was that existing works on efficient Transformers do not explicitly try to save activation memory, unlike our method. Although they can save memory not in terms of activation memory, we would like to mention that our method is unique and orthogonal to them.
> >
> > I would encourage the authors to include a discussion on this in revised version.
> >
> > **Regarding Lines 60-61:**
> > > It is the ratio of how much softmax is passed through among the layers that each input passes through inference in BERT and Transformer-base. It has nothing to do with the length of the sentence.
> >
> > Thank you for clarifying. Could the authors elaborate on details of ratio? Specifically, is the pairwise dot product, QK^T, part of the softmax operation, i.e., in the ratio's numerator? The attention module exhibits quadratic complexity in terms of the sequence length. Depending on this ratio's specifics, it may or may not be dependent on sequence length.
> >
> > **About Line 133 and Error Bound:**
> > >  Thus, the total amount of error given on the left-hand side in Equation 8 should not change regardless of the signs of each term. The reason we explicitly add the assumption is that we hope it would helps readers understand the process of the error-bound analysis. Although we believe the assumption does not affect the error bound, we will keep looking into it to see if there is a mistake or something we are missing.
> >
> > I understand the rationale behind adding the assumption for clarity in the error-bound analysis. However, while the total error on the left side of Equation 8 remains unchanged, without this assumption, moving summations internally becomes problematic (on the right side). I'm not convinced by the authors' belief that this assumption doesn't impact the error bound.
> >
> > My own attempt at the derivation, without this assumption, is as follows:
> > For any $i$, let $S_1$ be the set where $s_is_j > s'_is'_j$ and $S_2$ be the complementary set. We know $S_1 U S_2 = j \in \{1,2,\cdots n\}$. WLOG assume $s_is_i > s'_is'_i$.
> >
> > $
> > \sum_{j=1 \backslash i}^n
> > \left|s_i s_j-s_i^{\prime} s_j^{\prime}\right|
> > $
> >
> > $
> > = \sum_{j \in S_1, j \neq i} (s_i s_j-s_i^{\prime} s_j^{\prime}) -  \sum_{j \in S_2} (s_i s_j-s_i^{\prime} s_j^{\prime})
> > $
> >
> > $
> > = \sum_{j \in S_1, j \neq i} (s_i s_j-s_i^{\prime} s_j^{\prime}) -  \sum_{j \in S_2} (s_i s_j-s_i^{\prime} s_j^{\prime})
> > $
> >
> > $
> > = \sum_{j=1, j \neq i}^n (s_i s_j-s_i^{\prime} s_j^{\prime}) -  2 * \sum_{j \in S_2} (s_i s_j-s_i^{\prime} s_j^{\prime})
> > $
> >
> > $
> > = \left(s_i (1-s_i) -s_i^{\prime} (1-s_i')\right) -  2 * \sum_{j \in S_2} (s_i s_j-s_i^{\prime} s_j^{\prime})
> > $
> >
> > $
> > \leq |s_i (1-s_i) -s_i^{\prime} (1-s_i')| + \delta
> > $
> >
> > where $\delta = 2 * \sum_{j \in S_2} (s_i^{\prime} s_j^{\prime} - s_i s_j) > 0$
> >
> > Kindly clarify if I made a mistake or misunderstood something.
> >
> > While the empirical evidence presented is commendable, I urge the authors to either provide a proof devoid of this assumption or acknowledge its potential limitations in real-world scenarios.
> >
> > **Baselines:**
> > The paper seems to overlook key comparisons against pivotal baselines. In particular, the "Long Range Arena" is a well-known baseline employed in numerous efficient attention mechanisms. A comparison against this baseline would undoubtedly be beneficial.
> >
> > **Time-Memory Tradeoff:**
> > The authors' depiction of the time-memory tradeoff in the graphs is appreciated. However, a 3-6x overhead for even moderate sequence lengths, such as 400, is a significant overhead that would severely limit the use of proposed method.
> >
> > **Final Remarks:**
> > I carefully reviewed other feedback given significant difference between my own rating and others. Noting the novelty of the proposed method on approximating softmax, I have decided to adjust my score to 4. While the proposed method is novel, the paper currently falls short in terms of rigor. Specifically baseline comparisons, justification of bounds, and practicality due to overheads is preventing me from giving a higher rating.

---

> > > ### Author Response · Authors · 2023-08-18
> > >
> > > **Lines 286-290:**
> > > > Thanks for your comment. Your advice has allowed us to re-evaluate our method on its strengths and weaknesses against other methods. We will describe the differences between our method and other baseline methods.
> > >
> > > **Regarding Lines 60-61:**
> > > > The ratio calculation on lines 60-61 is simple but needs to be more specific. Our method does not consider the complexity of an attention module, which usually has a quadratic complexity depending on the sequence length. Instead, it is the ratio that takes into account only the number of layers without considering the size or dimensions of the inputs and outputs of each layer. For example, when the transformer's encoder has two embedding layers and one multi-head attention layer, the multi-head attention layer accounts for 33.3% (1/3). Also, in the case of $QK^T$, it is regarded as a single operation because it is the matrix multiplication of $Q$ and $K$. In this way, our ratio is presented to indicate how much the softmax in attention operation is used in an attention-based model in a layer-wise manner. We will clarify the way we calculate the ratio in our revised paper by describing the details of it.
> > >
> > > **About Line 133 and Error Bound:**
> > > > Appreciate your efforts in deriving the error bound. In a glimpse, your derivation seems correct, which we missed in our submission. We will double-check our analysis in light of your derivation and provide the correct error bound in our revised paper in the case that the assumption does not hold.
> > >
> > > **Baselines:**
> > > > We concur with your opinion that this paper needs more comparisons against many pivotal works you mentioned, especially 'Long Range Arena'. We will include thorough comparison discussions as well as experiments as much as we can in our revised paper in order to clarify the differences and similarities between our method and existing works.
> > >
> > > **Time-Memory Tradeoff:**
> > > > As mentioned in the memory trade-off analysis, the implementation of our method is not optimized in terms of execution time since it is written in naïve Pytorch (Python), not CUDA Kernel (C++), unlike the original softmax written in CUDA Kernel. Currently, we are working to reduce the time overhead by implementing our method with CUDA Kernel. Since the theoretical time overhead is just O(n log2 n), which is required for sorting, we believe that 3-6x time overhead will be dramatically decreased when the code optimization is done with some marginal time variations depending on the sequence length.
> > >
> > > **Final Remarks:**
> > > > Your thorough and pragmatic review has allowed us to better examine our method's practicality and limitation against other works. We will put effort into revising our work based on your constructive comments, especially including baseline comparisons, error-bound analysis, and time overhead reduction.

---

### Official Review · Reviewer_jkF6 · 2023-07-22

**Soundness:** 3 good
**Presentation:** 3 good
**Contribution:** 3 good
**Rating:** 5
**Confidence:** 3

**Summary:**

This paper is about approximating softmax function, by storing only a fraction of the entire softmax output in a memory.
The authors argue that by applying approximation on the softmax function, they were able to save memory usage of softmax activation up to 84%.
The motivation starts from the observation that the softmax activation module takes up a huge amount of memory consumption in widely-used Transformer-based models.
Instead of storing all softmax outputs, the work suggests storing only top-m highest and top-m lowest values of the outputs.
From error analysis, the optimality of storing only highest and lowest values is justified.
Using those stored fraction of values, the rest of values are approximated in the assumption that the values will follow a modified exponential distribution.
From experiments, the authors empirically prove that the proposed method has advantages of reducing memory usage while keeping performance degradation negligible.

**Strengths:**

- The proposed method is easy to apply.
- Experiments are done across multiple domains and tasks.
- Reducing the overhead of softmax has high impact on further research

**Weaknesses:**

- It's not clear how large or small the MAEs of ~0.05 (output error) or ~0.0003 (gradient error) are. Rather than mean absolute error, relative error would be more appropriate method to present the significance of error introduced by the approximation.
- Performance and memory usage comparison with other efficient softmax methods (e.g. LongFormer, Performer) would be useful.

(typo)

line 25: GTP-2 → GPT-2

**Questions:**

- In equation 5, the inverse of softmax to compute $z_i$ (i.e. input) from $s_i$ (i.e. output) depends on $\sum_{j=1}^n e^{z_j}$. Can you elaborate more on how $z_i$ are recovered from $s_i$? Does that imply that we don't need to care about $\sum_{j=1}^n e^{z_j}$, since it's partition function and the softmax function doesn't change output by transition?
- In equation 8, the first and the second term seem to be equivalent only when $s_i s_j \ge s_i' s_j'$ or $s_i s_j \le s_i' s_j'$ for all $i \neq j$. If the assumption does not hold, how does the error bound change?
- In Table 1, as far as I understood, $m$ indicates the number of values to be kept among highest and lowest values; in such case $2m$ values will be stored. But for $n=100$, it is written that memory usage is 50%; doesn't $m=50$ mean we save all output values?
- One of the old beliefs about softmax or attention mechanism is that having lower rank may suffer the performance. Is the proposed approximation method free from making softmax's rank lower?

**Limitations:**

I believe the limitations are addressed well in the checklist.

---

> ### Author Rebuttal · Authors · 2023-08-08
>
> We appreciate the time and effort you have dedicated to providing insightful feedback on our paper, including the typo checking.
>
> **Q1**  Inverse of softmax
> >We understand that Equation 5 may be confusing. In Equation 5, we can restore $z_i'$ (i.e., approximated softmax input) if we have $s_i'$ (i.e., approximated softmax output) and $\ln(\sum^n_{j=1} e^{z_j})$. In the forward execution, when the input goes through the softmax function, it stores the intact values of selected elements of the softmax output along with $\ln(\sum^n_{j=1} e^{z_j})$ in memory. Then, during the backward pass, by using $s_i'$ (i.e., approximated softmax output) and the stored $\ln(\sum^n_{j=1} e^{z_j})$, the input $z_i'$ is restored from Equation 5.
>
> **Q2** Error bound
> >As the reviewer correctly pointed out, the assumption may not always be held. However, we expect that the error bound does not change even if the assumption does not hold. That is because 1) the error bound is the sum of absolute error and 2) both the original softmax output ($s_i$) and the approximated output ($s'_i$) are positive numbers with a value between 0 and 1. Thus, the total amount of error given on the left-hand side in Equation 8 should not change regardless of the signs of each term. The reason we explicitly add the assumption is that we hope it would helps readers understand the process of the error-bound analysis. Although we believe the assumption does not affect the error bound, we will keep looking into it to see if there is a mistake or something we are missing. We appreciate again your careful and crucial comment on this.
>
> **Q3** $m=50$
> > Yes, as the reviewer mentioned, all the elements are stored in memory when $n=100$ and $m=50$ since it stores $2m$ elements in memory.
>
> **Q4** Lower rank
> > We appreciate the reviewer letting us consider an important aspect of softmax operation in the proposed method. We know about the low-rank performance degradation of the attention mechanism and softmax. Fortunately, the proposed method does not lower the rank of the softmax. It may seem to do so because the proposed approximation mechanism saves only $m$ selected softmax outputs in memory during the forward pass. However, the forward execution fully utilizes the full rank of the original softmax as a normal feedforward running, and then only the $m$ selected softmax outputs are stored in memory to be used later for the backward pass. During the backward pass, the non-stored softmax output elements are resorted to fully construct the exactly-same rank of the original softmax for gradient computation. Hence, the performance degradation caused by the lower rank of softmax is not the case that happens in the proposed method.
>
> **Gradient error**
> > We appreciate the reviewer’s advice regarding the readability and understandability of our manuscript. We agree with the reviewer that the current MAE metric may not help readers get the feel of the relative amount of error. So, as the reviewer suggested, we have converted the absolute gradient matrix error to the relative percentage error, i.e., [Epoch: 1 / RE: 7.70%], [Epoch: 8 / RE: 13.33%], [Epoch: 15 / RE: 8.28%], [Epoch: 20 / RE: 12.74%]. We will include the relative percentage error in our revised manuscript. In addition, through these improved indicators, we will try to further reduce the approximation error of the gradient matrix in future studies. Thanks again for your careful suggestion.
>
> We truly appreciate the time and effort you invested in reviewing our work and providing constructive feedback. Your contributions have had a positive impact on the quality of our research, and We are sincerely thankful for your support.

---

### Official Review · Reviewer_fWPe · 2023-08-03

**Soundness:** 3 good
**Presentation:** 3 good
**Contribution:** 3 good
**Rating:** 5
**Confidence:** 4

**Summary:**

This paper targets improving the memory efficiency of attention networks by reducing the activation storage of the softmax output. The authors propose to store only m highest and m lowest softmax output values, together with some auxiliary variables, and infer the missing part during backpropagation through interpolation. Results on several NLP tasks show encouraging results.


**Strengths:**

1. The idea is easy to follow and implement
2. The model obtains good performance and memory reduction on a set of tasks.

**Weaknesses:**

1. Important baselines are missing.
2. The author only reports memory consumption but doesn't show the effect on training time and the impact of larger model sizes.
3. Translation benchmarks should be improved.

**Questions:**

* There are many approaches also aiming to improve the efficiency of attention, such as FlushAttention and Checkpointing. As a reader, I expect to see how your method compares with them and whether they could be complementary to deliver further improvements.
* Apart from memory consumption, please provide training and inference time change.
* The authors only experiment with Transformer base models, but the objective of memory reduction is to enable larger models. The authors should explore how their approach performs as the model size increases substantially.
* For translation, a more convincing benchmark is WMT tasks. Examples in Multi30K follow simple patterns which artificially makes the translation task trivial. Please update your experiments with WMT benchmarks.

**Limitations:**

I didn't see particular limitations of their approach: the proposed method seems general to all softmax-based attention models.

---

> ### Author Rebuttal · Authors · 2023-08-08
>
> We thank the reviewer for providing valuable and positive comments regarding our work, and the questions that helped us to refine our work.
>
> **Q1** Comparison of our method with  FlashAttention and Checkpointing
> > Flash attention tries to speed up the attention mechanism by using different memory access rates for each memory layer on the GPU. In order to improve efficiency using different access speeds, it accelerates matmul and sofmtax, that is, memory-bound operations, in which memory access is greater than the amount of computation. For this purpose, tilling and recomputation are used. In the case of matrix matmul, it is efficiently changed using tilling, and in the case of softmax, it is made efficient through recomputation. In this case, in order to avoid using $O(n^2)$ memory required to recalculate the softmax, the input is reconstructed after saving the softmax normalization statistics. In this case, our method is less applicable as there is no storage or operation of the softmax output. If possible, we think that we can increase the storage efficiency by not storing the input and normalization statistics and applying our method. This possibility will definitely be confirmed and evaluated through further research. In contrast, we would like to mention the merits of our method. In our method, if there is a large language model that prioritizes performance over any efficiency, and if it is an attention-based model, our method can be used depending on the presence or absence of softmax for attention scoring, regardless of the structure. Also, in the case of flash attention, it is a hardware-dependent method, but our method is software-dependent and efficient through an algorithm, so there may be a difference. Of course, flash attention is already sufficiently efficient. The reason for this view is that Flash attention v2, which is written at the level of 'CUDA kernel', exists. To compare at the same level, our method will also have to be compared with efficiency at the level of the 'CUDA kernel'. To this end, we will continue to study this matter and revise our manuscript.
>
> > Checkpointing is a method of inference between the corresponding layer and the checkpointed state whenever a gradient is needed without storing the output and input values of some layers required in the backward process. This increases time complexity due to inference and reduces activation memory. A similarity to our method is that there is an additional process to find the unsaved value when the gradient is needed. The difference is that checkpointing is selectively stored according to the model and situation, and since our method can be used with any softmax function, there will be a difference in that there are no restrictions on the model. Furthermore, if we can restore some values to the state to be checkpointed, similar to our method, the efficiency of checkpointing will increase.
>
>
> **Q2** About time complexity
> > We have conducted a new experiment on the training time of our method. The result can be found in the Global rebuttal response. Since our current code implementation is written in slow and naïve "Pytorch" and "Python", we would like to point that it is not obvious and not fair to compare our method with "torch.nn.functional.softmax" optimized with the CUDA kernel. Given this situation, we observe that the training time increases when compared to the baseline. The detailed indicators are included in the Global rebuttal response. We will optimize the implementation of our algorithm through continuous code optimization, especially with the CUDA kernel, which we expect to bring a huge speed up.
>
> **Q3** Experiments with large models
> > We appreciate the reviewer’s constructive suggestion on the need for experiments on large language models. Although we experimented with XLNet (0.12 billion parameters), similar in size to the GPT-2 small (0.124 billion parameters), which showed competitive learning performance, we agree that additional experiments are needed to see if our method works effectively for much larger language models. We will conduct in-depth experiments on larger models as much as possible we can by utilizing all the sources we have and will include the experiment result in our revised manuscript. Before presenting the experiment results on larger models, here we can provide some hints on how our method will work when applied to larger models. An important aspect of our method is how many and which softmax output values should be stored in memory to effectively perform softmax approximation that does not affect the training of a model. In general, in the case of a larger model that needs to learn longer and presumably more difficult data (e.g., longer sentences), the number of stored softmax output values in memory will also get longer if the ratio of the stored softmax output values is maintained. Thus, although the data gets longer and possibly gets harder to learn, we can expect that the softmax output values would be effectively approximated with marginal errors, as more softmax output values are stored in memory proportionally to the data length. However, again, we will conduct experiments to see if this surmise is correct or not. It would be appreciated if the reviewer would bear with us on the experiment results.
>
> **Q4** WMT task experiment
> > Thanks for suggesting to experiment with WMT. As the reviewer requested, we have conducted a new experiment on "WMT2014 German-English" to validate the reliability of our method. The result shows that our method achieves the same model performance (perplexity) on the WMT task up to using 7.75 times less activation memory. The details of the experiment results are provided in the PDF file in the Global rebuttal response. We kindly ask the reviewer to take a look at it.

---

### Author Rebuttal · Authors · 2023-08-09

We want to extend our heartfelt thanks to all the reviewers for taking the time to review our research paper from diverse angles and offering constructive critiques. Your valuable insights have greatly enriched the quality of our research.

We have conducted new and re-experiments requested by the reviewers and included the results in the attached PDF file. We kindly ask the reviewers to check out the attached PDF file.

**Experiments on IMDb and WMT (Table 1 and Figure 1)**

>***Re-experiment with the IMDb task***
> >  The result on the left side of Table 1 is the re-evaluation of the IMDb task described in the paper. All experimental parameters are the same as in the paper. The sentence length is limited to 100 and the batch size is 128. The result using normal softmax (baseline) is 0.876. By applying our method, the value of the highest 1 and lowest 1 softmax elements out of 100 softmax elements are stored in memory ($m=1$), and it achieves an accuracy of 0.883 by using only 1.9% of softmax activation memory, which is slightly higher than the baseline. This variability seems to arise from subtle difficulties in approximating outputs from softmax inputs (from unstructured sequence data) that tend to deviate randomly from the exponential function, which we will further study.

> ***New experiment on the WMT-14 De-En task***
> > The right side of Table 1 is the new experiment result on Machine Translation learning using WMT-14 De-En data for the Transformer-base model. In the case of the WMT14 dataset, there are about 4 million train data and 3,000 validation and test data. We have experimented with WMT-14 De-En using around 20,000 train data samples due to the tight time limitation of the rebuttal. The result shows the perplexity of the test dataset every $m$. The result using normal softmax (baseline) is 1.004. With our method, the same perplexity (1.004) is obtained by storing the values of the highest 2 and lowest 2 softmax outputs out of 100 elements. Figure 1 shows the learning performance trajectory with various $m$ over training epochs, which shows that our method draws almost the same trajectory as the baseline (normal).


**Experiments on time and memory complexity (Figure 2 and Figure 3)**

> ***Time and memory evaluation according to sentence length and batch size at a single softmax layer***
> > Figure 2(a) shows the activation memory usage and time complexity of the original softmax function and the softmax function to which our method is applied, as described in the caption. It can be observed that the time complexity changes according to the sequence length regardless of the number of softmax elements stored in memory. We can also observe that the time increases rapidly from a minimum of 4$\times$ to about 30$\times$ due to inefficient code implementations in the process of sorting, cutting, and rearranging the matrix, which are required by the proposed softmax output approximation. This is because we used a non-optimal way to implement our algorithm (Python, not CUDA kernel), unlike the original softmax function highly optimized with CUDA kernel. This will continue to be modified in our work, and we plan to implement it with the highly optimized CUDA kernel as the final goal.


> ***Evaluation of the effect of batch size and sequence length on the turnaround time of our method***
> > Figure 2(b), we present the factors that increase the time complexity when our method is applied. Each x-axis is batch size, and each line represents sentence length. As we can see, the batch size changing the softmax output shape to $n$ has a more significant effect on the inference speed than the sequence length ($n^2$). Based on this experiment result, we will optimize our code and make it faster.

> ***Time complexity evaluation of the entire model***
>> Figure 3, we proceed to the time evaluation spent by all the layers of the Transformer-base model, unlike Figure 2, which measures the amount of time spent at a single softmax layer. In this experiment, three different batch sizes of 32, 64, and 128 are used, which are divided into two groups; blue bars (original softmax; baseline) and green bars (our method). For the x-axis of $n=100$, the increase in time complexity due to our method occurs on average by a factor of about  1.7$\times$. In the case of $n = 200$, the complexity increases by about 5 $\times$ on average, and the amount of time required is about 6.3 $\times$ at most, which is smaller than the amount of time spent at a single softmax layer. Also, the trend of linear change is observed with the batch size and sequence length. Please note that these results may vary depending on the situation of the GPU or CPU. However, as we mentioned, we will improve our implementation of the proposed softmax output approximation method and will evaluate the time overhead again with the optimized code that can execute our algorithm with more efficiency while the significant memory-saving capability is maintained.

---

### Decision · Program_Chairs · 2023-09-21

**Decision:**

Accept (poster)

**Comment:**

Strong paper on softmax output approximation to reduce memory requirements. Most of the reviewers think this paper should be accepted. Reviewer UEEi has some concerns but I think the authors have sufficiently addressed them in the rebuttal. I recommend acceptance.